# Flash-DMD: Unifying Distillation and Refinement for High-Fidelity Few-Step Image Generation

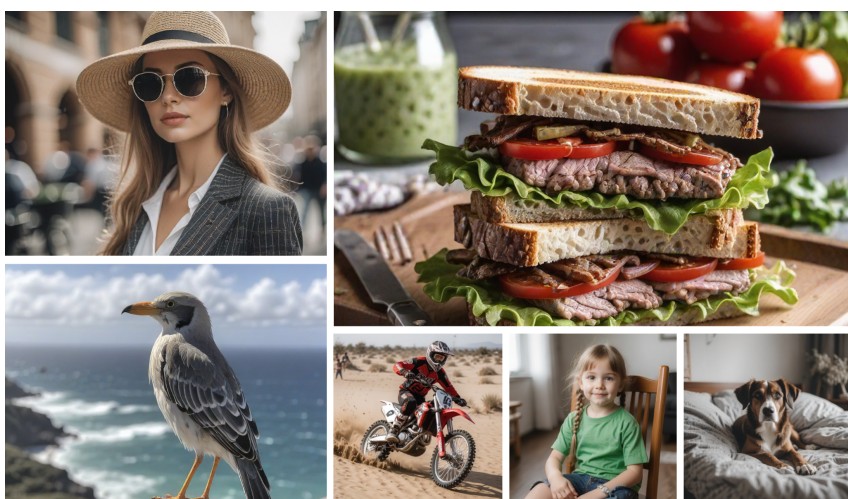

Figure 1: Samples from Flash-DMD, taking less than 3% training cost of DMD2.

## Abstract

Diffusion Models have emerged as a leading class of generative models, yet their iterative sampling process remains computationally expensive. Timestep distillation is a promising technique to accelerate generation, but it often requires extensive training and leads to a degradation in image quality. Furthermore, fine-tuning these distilled models to optimize for specific objectives, such as aesthetic appeal or user preference, using Reinforcement Learning (RL) is notoriously unstable, and easily falls into reward hacking. In this work, we introduce Flash-DMD, a novel framework that enables fast convergence with distillation and stable RL-based refinement for stable optimization. Specifically, we first propose an efficient timestep-aware distillation strategy that siginicantly reduce training cost with enhanced human preference and realism. Second, and most critically, we introduce a joint training scheme where the model is fine-tuned with an RL objective while the timestep distillation training continues simultaneously. We demonstrate that the stable, well-defined loss from the ongoing distillation acts as a powerful regularizer, effectively stabilizing the RL training process and preventing policy collapse. Our experiments show that our proposed Flash-DMD not only converges significantly faster but also achieves state-of-the-art generation quality in the 4-step sampling regime, outperforming existing methods in human preference evaluations. Our work presents an effective paradigm for training efficient, high-fidelity, and stable generative models. Codes are attached in the supplementary.

## 1 Introduction

Diffusion models (Ho et al., 2020; Rombach et al., 2022; Podell et al., 2023; Esser et al., 2024; Labs, 2024) have demonstrated remarkable success in text-to-image generation in recent years. However,

generating high-quality images necessitates numerous iterative denoising steps, resulting in substantial computational overhead, especially as models grow in size and complexity. This overhead poses a significant obstacle to real-time or resource-constrained deployment. To address this issue, various diffusion distillation techniques have been developed to distill multi-step teacher diffusion models into efficient student models that can produce comparable image quality in just one or a few inference steps (Luo et al., 2023a; Wang et al., 2024; Yin et al., 2024c;b; Lin et al., 2024; Chadebec et al., 2025; Ge et al., 2025; Lu et al., 2025). However, existing distillation methods are often inefficient and resource-intensive, requiring thousands of GPU hours for training. This significantly limits its accessibility to research groups and institutions with limited resources, and hinders rapid iteration and deployment in practical applications.

Among existing distillation methods, Distribution Matching Distillation (DMD) methods (Yin et al., 2024c;b; Lu et al., 2025; Ge et al., 2025) stand out for their superior generative quality, leveraging variational score distillation objectives (Yu et al., 2023) to align the output distributions of student and teacher models. However, this objective function suffers from unstable training and a tendency to mode seeking. Some approaches have employed adversarial methods to mitigate these problems. DMD2 (Yin et al., 2024b) proposes latent adversarial regulations with real images and designs a Two-Time scale Update Rule (TTUR) to stabilize training, but it combines the GAN (Sauer et al., 2024b;a) framework with DMD in a naive manner, neglecting the timestep aware feature of timestep distilled diffusion models, and the fake score $\mu_{fake}$ is trained both to discriminate real and generated images, but also to track the distribution changes of student models. These design compromises its efficiency in matching the distribution of teacher models. Furthermore, DM loss is inefficient in the latter part of distillation as it is hard to guide detailed learning, preventing it from effectively guiding the student diffusion model. These observations motivate our core research questions:

> **Q1** *In the early phase, how can we more effectively coordinate distribution matching with perceptual realism enhancement to accelerate convergence and stabilize training?*
> **Q2** *In the later phase, how can we refine the student model for better visual details and perceptual fidelity when gradient signals from distribution matching become less informative?*

To address the inefficiencies of the distribution matching methods, we proposed Flash-DMD, a twofold method in the four-step distillation task that outperforms DMD2 in the human preference benchmark with a much smaller training cost, while preserving superior perceptual realism. Specifically, our Flash-DMDfollows different principles in the early and later generation phases.

*In the early phase*, the denoising performance at different stages varied, so the distillation target should also differ. We accordingly decoupled the GAN and distribution matching frameworks. At high-noise timesteps, the denoising model's primary objective is to learn global composition and structure from the teacher, and DM loss is effective to process noisy latents. Therefore, we use a pure DM loss to align the student model with the teacher model's output distribution. At low-noise timesteps, the model focuses on refining fine-grained details and enhancing perceptual realism. Thus, we use an adversarial loss to match the distribution of real images and improve the photorealism of the final outputs.

*In the later phase*, as gradient signals from distribution matching diminish, we directly optimize for visual quality and human preference. Our approach combines the DMD framework with latent reinforcement learning to efficiently refine the model's handling of fine-grained details. Simultaneously, the distribution matching (DM) loss constrains domain shift, preventing the mode collapse and visual degeneration common in few-step reinforcement learning. This ensures the final results maintain high realism, avoiding reward hacking and the emergence of "oil painting" artifacts.

By combining faster convergence in the early phase with finer optimisation in the latter phase, we demonstrate the efficiency and superior performance of our method of distilling from SDXL to produce high-quality, realistic images. Notably, our method achieves the highest human preference scores while requiring the lowest training cost to date.

To summarize, our main contributions are threefold:

- We introduce Flash-DMD, a highly efficient framework for Distribution Matching Distillation. we decouple training objectives via a timestep-aware strategy to efficiently distill the fundamental distribution of the teacher model in low-SNR timesteps and refine perceptual quality and texture in high-SNR timesteps, and we counteract the mode-seeking of the

DM loss with a SAM-based Pixel-GAN that robustly enhances realism. The combination of these strategies and the stabilized score estimator allows for fastest convergence and stabilized distillation.

- Extensive experiments demonstrate that our method achieves superior performance compared to both the teacher model and baseline in human preference metrics, using only $2.1\%$ of the training cost of DMD2. Further improvements are attained with an additional $6.2\%$–$35.4\%$ of DMD2's training cost with even enhanced realism and textural detail.

- We successfully integrate reinforcement learning into the distillation process, creating a unified framework that seamlessly combines post-training refinement with distillation. This innovation eliminates the need for separate reinforcement and distillation phases, significantly reducing computational training costs while enabling efficient optimization of fine-grained details and perceptual fidelity in few-step image generation.

## 2 RELATED WORK

**Diffusion Distillation.** *Progressive Distillation* (Salimans & Ho, 2022; Meng et al., 2023; Lin et al., 2024) reduce inference steps in diffusion models by iteratively halving them, ultimately producing a one-step generator. Although effective, this iterative process is computationally expensive and is constrained by the preceding teacher model's quality, leading to compounding errors. *Consistency Distillation* (Luo et al., 2023a;b) enforces a consistency constraint for the diffusion models, stipulating that any point on a given trajectory will revert to its starting point. However, it leads to performance degradation in the few-step inference. To migrate the issue, recent works (Wang et al., 2024; Ren et al., 2024; Zheng et al., 2024) have segmented the trajectory and progressively perform distillation on timestep segments, thereby achieving enhanced stability and fidelity. *Adversarial Distillation* introduces a discriminator to align the few-step student's output with the multi-step teacher's, either at the pixel level (Sauer et al., 2024b) or latent level (Sauer et al., 2024a), DMD2(Yin et al., 2024b). DMD2(Yin et al., 2024b) also uses latent adversarial training to match real-world data distribution, but its straightforward combination of adversarial loss and distribution matching may introduce conflicting objectives that can hinder overall distillation efficiency. *Score Distillation* was subsequently adapted for the distillation of diffusion models themselves (Yin et al., 2024c; Nguyen & Tran, 2024; Franceschi et al., 2023). An early approach, Distribution Matching Distillation (DMD) (Yin et al., 2024c), aims to minimize the KL-divergence between the teacher and student distributions. DMD2 (Yin et al., 2024b) replaced regression loss with adversarial loss for better realism. Building on this, Adversarial Distribution Matching (ADM) (Lu et al., 2025) introduced a GAN framework with Hinge loss, while SenseFlow (Ge et al., 2025) optimized scorers and discriminators for efficient distillation of larger models.

**Reinforcement Learning in T2I Generation.** Reinforcement learning is rapidly migrating to image generation tasks to align large-scale diffusion models with human feedback. Direct Preference Optimization (DPO) (Wallace et al., 2024; Liang et al., 2025; Lee et al., 2025; Li et al., 2025b; Miao et al., 2025; Zhang et al., 2025a) and Group Relative Policy Optimization (GRPO)(Liu et al., 2025; Xue et al., 2025; Li et al., 2025a; Wang et al., 2025; He et al., 2025) are two popular paradigms. The former methods construct offline or online win-lose pairs and back-propagate the preference order by the Bradley-Terry formed objectives. The latter methods sample a group of images on the SDE/mixed ODE-SDE trajectory, calculate the normalized advantage within the group, and constrain the policy generation direction. However, current research on performing RL on few-step models remains quite limited. Miao et al. (2025) proposed pairwise sample optimization (PSO), strengthening the relative likelihood margin between the training and reference sets. In contrast to PSO, we leverage a reward model operating in the latent space without relying on extrinsic positive targets(e.g., human-preferred labels, stylized images), which exclusively utilizes self-generated image pairs produced by the model itself and reduces computational overhead during training.

## 3 METHODOLOGY

### 3.1 PRELIMINARY OF DISTRIBUTION MATCHING DISTILLATION

Given pretrained diffusion model $\mathcal{T}_\phi(x_t, t)$ as teacher model, where $x_t$ is noisy sample at timestep $t \sim \mathcal{U}(\mathbf{1}, \mathbf{T})$, DMD(Yin et al., 2024c) and DMD2(Yin et al., 2024b) distill it into few-step effi-

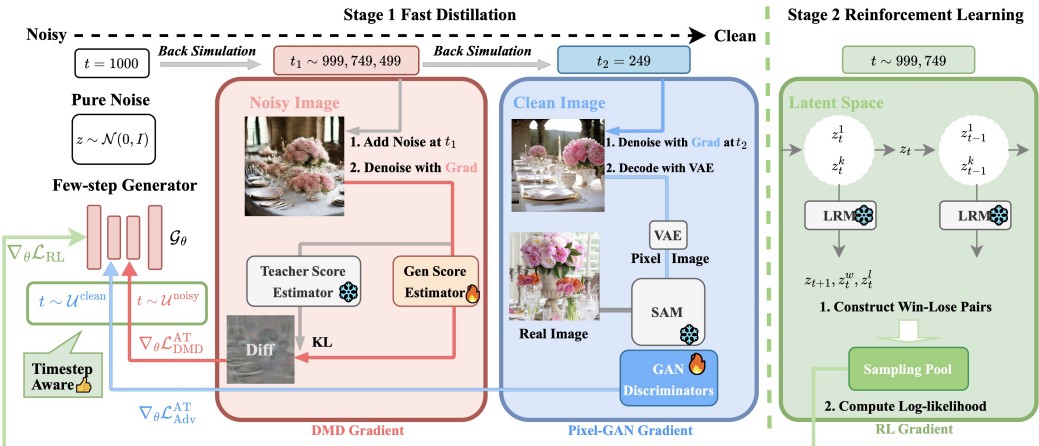

Figure 2: Framework of our proposed method Flash-DMD. There are two training stages: fast distillation and reinforcement learning. In stage 1, we take timestep-aware strategy and PixelGAN to efficiently create a realistic version of the teacher. In stage 2, latent preference optimization on high noise timesteps is utilized to enhance generation fidelity to surpass the teacher.

cient generator $\mathcal{G}_\theta(x_t, t)$ by minimizing the reverse KL divergence between the teacher model's distribution $p_{\text{tea}}$ and the few-step generator's distribution $p_{\text{gen}}$. DMD series methods estimate $p_{\text{tea}}$ through score estimator $\mu_{\text{tea}}(x_t, t)$, and $p_{\text{gen}}$ is tracked with estimator $\mu_{\text{gen}}(x_t, t)$. Score function of the diffused distribution is:

$$s_{\text{gen}}(x_t, t) = -\frac{x_t - \alpha_t \mu_{\text{gen}}(x_t, t)}{\sigma_t^2}; \, s_{\text{tea}}(x_t, t) = -\frac{x_t - \alpha_t \mu_{\text{tea}}(x_t, t)}{\sigma_t^2} \tag{1}$$

where $\alpha_t, \sigma_t > 0$ are scalars determined by the noise schedule. $s_{\text{gen}}$ and $s_{\text{tea}}$ are vector fields that point towards higher density of distribution. Gradient of Distribution Matching objective w.r.t. $\theta$ is,

$$\nabla_\theta \mathcal{L}_{\text{DMD}} = \mathbb{E}_{z,t} \left[ -\left( s_{\text{tea}}(\mathcal{G}_\theta(z, t)) - s_{\text{gen}}(\mathcal{G}_\theta(z, t)) \right) \frac{d\mathcal{G}_\theta(z, t)}{d\theta} \right], \tag{2}$$

where $z \sim \mathcal{N}(\mathbf{0}, \mathbf{I})$, $t \sim \mathcal{U}(\mathbf{0}, \mathbf{T})$. In addition to the Distribution Matching objective, DMD2 introduces the combination with adversarial training with real images. Gradient of generator's adversarial objective w.r.t. $\theta$ is,

$$\nabla_\theta \mathcal{L}_{\text{AdvGen}} = \mathbb{E}_{z,t} \left[ \log \mathcal{D}\left( \mathcal{G}_\theta(z, t) \right) \frac{d\mathcal{G}_\theta(z, t)}{d\theta} \right], \tag{3}$$

where $\mathcal{D}$ is the discrimator forward process. The score estimator of teacher $\mathcal{T}_\phi(\cdot)$ is itself, the generator's score estimator $\mu_{\text{gen}}^\psi(\cdot)$ is initialized with $\mathcal{T}_\phi(\cdot)$, and is dynamiclly updated to track $p_{\text{gen}}$ with diffusion loss:

$$\mathcal{L}_{\text{Diffusion}} = \mathbb{E}_{x_{t-1}, t, \epsilon \sim \mathcal{N}(\mathbf{0}, \mathbf{I})} [\| \mu_{\text{gen}}^\psi(x_t, t) - \epsilon \|_2^2], \tag{4}$$

DMD2 reuses the parameter $\psi$ of $\mu_{\text{gen}}^\psi(x_t, t)$ and extra trainable heads to distinguish $p_{\text{gen}}$ and real image distribution $p_{\text{real}}$. Gradient of estimators's adversarial objective w.r.t. $\psi$ is,

$$\nabla_\psi \mathcal{L}_{\text{AdvDisc}} = \mathbb{E}_{z,t,x_{\text{real}} \sim p_{\text{real}}} \left[ \log \mathcal{D}\left( x_{\text{real}} \right) \frac{dD\left( x_{\text{real}} \right)}{d\psi} - \log \mathcal{D}\left( \mathcal{G}_\theta(z, t) \right) \frac{dD\left( \mathcal{G}_\theta(z, t) \right)}{d\psi} \right], \tag{5}$$

## 3.2 TRAINING INEFFICIENCY OF DMD SERIES

Despite their impressive performance, methods in the DMD series are characterized by significant computational overhead during distillation. This is evident in the extensive training schedules required by prominent models. For instance, the original DMD(Yin et al., 2024c) required $20,000$ iterations with a batch size of 2,304 to distill Stable Diffusion v1.5 (Rombach et al., 2022) for single-step generation. Similarly, DMD2 (Yin et al., 2024b) used $24,000$ iterations to distill SDXL (Podell et al., 2023) for four-step generation, and ADM (Lu et al., 2025) used $16,000$ iterations

for single-step SDXL distillation. Given the strong empirical results and open-source implementation of DMD2, we select it as the foundation for our investigation into these inefficiencies. One primary source of inefficiency in DMD2 stems from its optimization strategy. As noted by Cheng et al. (2025), DMD2 simultaneously optimizes the model using two distinct gradients: a distribution-matching gradient (Eq. equation 2) and an adversarial gradient (Eq. equation 3). A direct summation of these gradients can introduce conflicting objectives, potentially steering the model toward a suboptimal state. This conflict can degrade both the accuracy of the distribution matching and the perceptual quality of the generated images, thereby hindering efficient convergence. A second challenge lies in the dual role assigned to the generator's score estimator. It is tasked with two demanding objectives: tracking the output distribution of $\mathcal{G}_\theta(\cdot)$(Eq. equation 4) and discriminating between real and generated samples (Eq. equation 5). To stabilize this complex dynamic, a two-time scale update rule (TTUR) is employed in DMD2, where score estimator is updated five times for every single update of the generator $\mathcal{G}_\theta(\cdot)$. This significantly contributes to the model's overall training inefficiency.

### 3.3 Faster Convergence in early phase

**Adversarial Training is Necessary.** DMD2 framework optimizes the generator by naively summing the Distribution-Matching (DM) loss from the teacher and an adversarial loss against real images at every timestep. This superposition of gradients can result in suboptimal and inefficient optimization. When we remove the adversarial teacher entirely, we observe that under pure DM loss supervision, the generator rapidly converges to a suboptimal domain, producing outputs with unnaturally high contrast and lacking fine-grained textures. We attribute this behavior to the mode-seeking nature of the reverse KL divergence, an observation also discussed in ADM (Lu et al., 2025). This finding underscores the necessity of the adversarial loss with real images for perceptual fidelity.

**Decoupling Losses with a Timestep-Aware Strategy.** We observe that the generator's objective changes throughout the denoising process. For a 4-step distilled model, the initial, high-noise timesteps (low Signal-to-Noise Ratio, or SNR) primarily establish global composition and structure, and the low-noise timesteps (high SNR) focus on refining details, textures, and color tones to enhance realism. This observation is corroborated by findings of Cheng et al. (2025) in video generation tasks, which noted that the adversarial training in DMD2 is most active at high SNRs, whereas DM loss excels at guiding the model through high-noise regimes. Based on these insights, we assign DM loss and adversarial loss to distinct timesteps: *during the high-noise regime* (steps 1-3 in a 4-step model), we optimize the generator exclusively with the DM loss (Eq. equation 2). This allows the model to efficiently learn the teacher's fundamental distribution and ODE trajectory in the early phases of generation. *For the low-noise step*, we apply the adversarial loss against real images, enabling the model to refine perceptual quality and texture in the final denoising step.

During each generator update, we sample one timestep $t \sim 999, 749, 499, 249$ and $x_t$ from DMD2's back-simulation forward process $\mathcal{B}$ to compute the DM loss:

$$x_t = \texttt{Detach}(\mathcal{B}(z,t)); \nabla_\theta \mathcal{L}_{\text{DMD}}^{\text{AT}} = \mathbb{E}_{z,t}\left[-\left(s_{\text{tea}}(\mathcal{G}_\theta(x_t,t)) - s_{\text{gen}}(\mathcal{G}_\theta(x_t,t))\right)\frac{d\mathcal{G}_\theta(x_t,t)}{d\theta}\right], \quad (6)$$

then employ $\mathcal{B}$ to propagate the denoised output $x_{t-1}$ to a final clean image$x_0$:

$$x_{t_1} = \mathcal{G}_\theta(x_t,t); x_0 = \texttt{Detach}(\mathcal{B}(x_{t_1},0)) \qquad (7)$$

where $\texttt{Detach}$ denotes stop gradient, $x_0$ is then used for the adversarial loss computation. We perform diffusion forward on $x_0$ to obtain the noisy sample $x_{249}$. Gradient for adversarial loss is:

$$\nabla_\theta \mathcal{L}_{\text{AdvGen}}^{\text{TA}} = \left[\mathbb{E}_{t=249}\log\mathcal{D}\left(\texttt{Deocde}\left(\mathcal{G}_\theta(x_{249},t)\right)\right)\frac{d\mathcal{G}_\theta(x_{249},t)}{d\theta}\right] \qquad (8)$$

where the $\mathcal{D}(\cdot)$ is the pixel-level discriminator, and $\texttt{Decode}$ is the decode process of SDXL-VAE(Podell et al., 2023). This timestep-aware strategy reduces interference between these two optimization objectives. Our experiments demonstrate that this approach significantly improves training efficiency while generating high-quality images with enhanced realism and textural detail.

**Pixel-GAN alleviates mode-seeking.** In order to enforce realism and structural coherence, the present study performs adversarial learning directly in the pixel space using a discriminator based on features that is new and innovative. In contrast to a conventional latent-space GAN, the discriminator in our model is constructed upon the frozen vision encoder of the Segment Anything Model

(SAM) to extract hierarchical features with multiple trainable discriminator heads attached. The discriminator's trainable parameters $\omega$ are updated via the Hinge loss:

$$\mathcal{L}_{\text{AdvDisc}}^{\text{PG}} = \mathbb{E}_{x_{\text{real}}} \left[ - \log \mathcal{D}_\omega \left( x_{\text{real}} \right) \right] + \mathbb{E}_{z, \mathbf{t=249}} \left[ \log \mathcal{D}_\omega \left( \text{Decode}(\mathcal{G}_\theta(x_t, t)) \right) \right] \tag{9}$$

The discriminator is characterized by its exceptional sensitivity to local geometric structures and fine-grained textures, a capability that is facilitated by SAM's powerful, general-purpose representations, as noted by Lu et al. (2025). This pixel-level supervision exerts a stringent realism constraint from the training's earliest stages, compelling the generator to expeditiously discern and anchor to diverse, high-fidelity modes within the data distribution. This approach effectively prevents premature convergence to simplistic or blurry solutions (mode-seeking).

**Stabilize Score Estimator.** In contrast to DMD2 Yin et al. (2024b), where the score estimator is required to serve as a discriminator (Eq. equation 5) and thus faces conflicting optimization objectives, our approach trains $\mu_{\text{gen}}^\psi$ solely via the diffusion loss (Eq. equation 4), eliminating the tasking burden and training complexity for $\mathcal{G}_\theta(\cdot)$. Our experiments show that updating the score estimator only once or twice per generator update (TTUR=1,2) is sufficient for stable and accurate distribution tracking. This lightweight coupling leads to more stable training dynamics and superior sample fidelity compared to DMD2 with TTUR=5, while reducing computational overhead.

Similar to implicit distribution alignment proposed by Ge et al. (2025), we also adopt an Exponential Moving Average (EMA) update strategy to ensure that the score estimator $\mu_{\text{gen}}^\psi$ accurately tracks the evolving distribution of the generator $\mathcal{G}_\theta(\cdot)$. Specifically, after each generator update, we inject the latest generator parameters into the score estimator using an EMA coefficient $\lambda_{\text{ema}}$, i.e.,

$$\psi \leftarrow \lambda_{\text{ema}} \psi + (1 - \lambda_{\text{ema}})\theta, \tag{10}$$

which enables $\mu_{\text{gen}}^\psi$ to closely follow the generator's trajectory with minimal additional updates.

**Putting everything together.** We introduce Flash-DMD, a highly efficient framework for Distribution Matching Distillation. In summary, Flash-DMD's training objectives via a timestep-aware strategy efficiently distill the fundamental distribution of the teacher model in low-SNR timesteps and refine perceptual quality and texture in the final high-SNR timestep. To counteract the mode-seeking tendency of the DM loss, we introduce a SAM-based Pixel-GAN that robustly enhances realism. The combination of these strategies and the stabilized score estimator enables a more effective and balanced optimization of the generator $\mathcal{G}_\theta(\cdot)$.

### 3.4 REINFORCEMENT LEARNING FOR FINER PERFORMANCE IN THE LATER PHASE

**Reinforcement learning boosts model performance.** Leveraging the above training paradigm, we have developed a student generator capable of rivaling the teacher model. Subsequently, our focus shifts to enhancing its performance beyond that of the teacher and deploying it in practical scenarios. Preference optimization provides a direct shortcut to significantly improve image fidelity, aesthetics, and detail richness with limited resources.

**Choosing a suitable Reward Model.** Selecting an appropriate reward model is a critical step. We begin with an empirical investigation of the prevalent models and systematically summarize their characteristics in the appendix. The empirical findings lead to the following conclusions: 1) Most models operate in pixel space, with only the Latent Reward Model (LRM) exploring in latent space. 2) Although the timestep constitutes a critical factor for text-to-image models, it is neglected by most reward models, with a few exceptions. 3) In general, these reward models score on dimensions such as image fidelity, aesthetics, and text-image alignment. In light of the analysis above, we incorporate LRM into our distillation training pipeline.

**The best of both worlds.** Born from stable diffusion models, LRM can score noisy latent representations at any timestep without the need for complex VAE decoder transformations. This capability inherently meets our demands. A subsequent key challenge is how to integrate LRM into our framework to achieve greater gains effectively. To adapt the approach to our method, we introduce several key changes: 1) In the sampling process for win-lose pair construction and log-likelihood computation stage, we replace the original noise scheduler, Denoising Diffusion Implicit Model (DDIM), with Latent Consistency Model (LCM) since the distilled model is compatible with the latter. 2) As shown in Fig. 3, with the same initialized noise, it is evident that images sampled at high-noise steps exhibit better diversity in layout and fine-grained details compared to those from

low-noise steps. As a result, we only perform stochastic sampling in the high-noise phase, altering latent representations that are deemed optimal/suboptimal by the reward model. 3) The dynamic threshold is removed because the original dynamic boundary range is insufficient to adapt to the distribution characteristics of distilled models adequately. 4) We combine the logarithmic likelihood loss with the vanilla loss of Flash-DMDduring the training process rather than applying preference optimization separately. We will explore a reasonable time scale update rule in the appendix file.

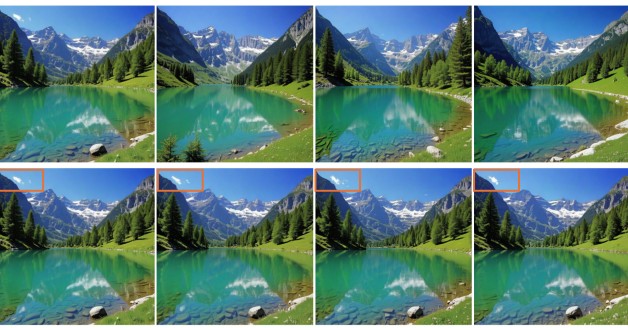

Prompt: Designs Similar to Lake Verde In The Alps

Figure 3: Sampling variance analysis at different time steps. The first row displays samples obtained at the 999th denoising step, while the second row corresponds to the 499th step.

**Win-lose Pair Construction and Final RL Loss.** Given a generator $\mathcal{G}_\theta(\cdot)$ distilled from a pretrained diffusion model $\mathcal{T}_\phi(\cdot)$, it can sample clean images from pure noise $z_T \sim \mathcal{N}(\mathbf{0}, \mathbf{I})$, conditioned on text prompt $c$, within $T = 4$ steps. At high-noise timesteps, we sample a set of $k$ noisy latent images $\{z_{t-1}^1, ..., z_{t-1}^k\}$ from the same initial latent image $z_t$. LRM predicts preference scores. The samples corresponding to the highest and lowest normalized scores are selected as winlose pairs, thereby constructing paired training data $(z_t, z_{t-1}^w, z_{t-1}^l)$ to the sampling pool. These pairs are subsequently used to minimize the loss function:

$$\mathcal{L}_{rl} = -\mathbb{E}_{z_{t-1}^w, z_{t-1}^l \sim p_\theta(z_{t-1}|z_t, c)}[\log \sigma(\beta \log(\frac{p_\theta(z_{t-1}^w|z_t, c)}{p_{ref}(z_{t-1}^w|z_t, c)}) - \beta \log(\frac{p_\theta(z_{t-1}^l|z_t, c)}{p_{ref}(z_{t-1}^l|z_t, c)}))] \quad (11)$$

where $p_\theta(z_{t-1}|z_t, c)$ denotes the backward process to denoise $z_t$ and can be formulated as follows:

$$p(z_{t-1}|z_t) = \mathcal{N}(\mu_t, \sigma_t^2 \epsilon_t), \epsilon_t \sim \mathcal{N}(0, I) \quad (12)$$

$$\mu_t = \sqrt{\overline{\alpha}_{t-1}} \cdot \hat{z}_{0,t} + \sqrt{1 - \overline{\alpha}_{t-1} - \sigma_t^2} \cdot \epsilon_\theta(z_t, t) \quad (13)$$

$$\hat{z}_{0,t} = c_{out}(t) \cdot (\frac{z_t - \sqrt{1 - \overline{\alpha}_t} \cdot \epsilon_\theta(z_t|c)}{\sqrt{\overline{\alpha}_t}}) + c_{skip}(t) \cdot z_t \quad (14)$$

where $\hat{z}_{0,t}$ denotes clean latent images predicted by noise predictor $\epsilon_\theta(\cdot)$ at timestep t. $c_{skip}(\cdot)$ and $c_{out}(\cdot)$ are differentiable functions predefined in the LCM scheduler.

## 4 EXPERIMENTS

### 4.1 IMPLEMENTATION DETAILS

**Experiment Setup** For the first phase, we utilize a filtered set from the LAION 5B (Schuhmann et al., 2022) dataset to provide high-quality image-text pairs for training, following the setting of (Yin et al., 2024a). For discriminator conditioning, we adopt the vision encoder from Kirillov et al. (2023) as the backbone to extract image representations. The structure of trainable discriminator heads follows the 2D architecture of (Lu et al., 2025). For the second phase, we adopt the training dataset from the first phase and utilize the Latent Reward Model from (Zhang et al., 2025a). We sample a set of noisy latent images at the high-noise timesteps $t = 749, 999$ and set $k = 4$ following (Zhang et al., 2025a). We conduct all of our experiments on NVIDIA H20 GPUs.

Table 1: Comparison with other distillation methods on COCO-10k dataset.

| Method | #NFE | ImageReward ↑ | CLIP ↑ | Pick ↑ | HPSv2 ↑ | MPS ↑ | Cost ↓ |
|---|---|---|---|---|---|---|---|
| SDXL | 100 | 0.7143 | 0.3295 | 0.2265 | 0.2865 | 11.87 | - |
| LCM-SDXL | 4 | 0.5562 | 0.3250 | 0.2236 | 0.2818 | 11.11 | - |
| SDXL-Lightning | 4 | 0.6952 | 0.3268 | 0.2285 | 0.2888 | 12.15 | - |
| SDXL-Turbo | 4 | 0.8338 | 0.3302 | 0.2286 | 0.2899 | 12.25 | - |
| NitroSD-Realism | 4 | 0.9112 | 0.3274 | 0.2291 | 0.2975 | 12.43 | - |
| NitroSD-Vibrant | 4 | 0.8419 | 0.3201 | 0.2205 | 0.2865 | 11.13 | - |
| DMD2-SDXL | 4 | 0.8748 | **0.3302** | 0.2309 | 0.2937 | 12.41 | 128*24k |
| **Flash-DMD under Phase 1** | | | | | | | |
| TTUR1-1k | 4 | 0.9509 | 0.3292 | 0.2322 | 0.2968 | 12.67 | **64*1k (2.1%)** |
| TTUR2-4k | 4 | 0.9450 | 0.3291 | 0.2322 | 0.2969 | 12.65 | 64*4k (8.3%) |
| TTUR2-8k | 4 | **0.9740** | 0.3298 | **0.2327** | 0.2981 | **12.71** | 64*8k (16.7%) |
| TTUR5-18k | 4 | 0.9426 | **0.3302** | 0.2319 | **0.2982** | 12.63 | 64*18k (37.5%) |

Table 2: Comparison with other methods with reinforcement learning on COCO-10k dataset.

| Method | #NFE | ImageReward ↑ | CLIP ↑ | Pick ↑ | HPSv2 ↑ | MPS ↑ | GPU Hours ↓ |
|---|---|---|---|---|---|---|---|
| Hyper-SDXL | 4 | **1.085** | 0.3300 | 0.2324 | **0.3030** | 12.45 | 400 A100 |
| PSO-DMD2 | 4 | 0.9157 | 0.3285 | 0.2338 | 0.2897 | 12.53 | 160 A100 |
| LPO-SDXL | 40 | 1.042 | **0.3324** | 0.2342 | 0.2965 | 12.58 | 92 A100 |
| Flash-DMD | 4 | 1.004 | 0.3285 | **0.2346** | 0.2930 | **12.84** | **12 H20** |

**Evaluation Tasks and Baseline**    The evaluation of image generators is conducted on 10K prompts from COCO 2014 (Lin et al., 2014), adhering to the DMD2(Yin et al., 2024a) framework, containing 10,000 images. We present the result of CLIP score(Radford et al., 2021) (ViT-B/32) to evaluate text-image similarity, and we adopt a set of advanced preference-based metrics to thoroughly evaluate the quality of generated images from multiple human-aligned perspectives. We use HPSv2(Wu et al., 2023) to measure fine-grained image-text semantic alignment, focusing on how well the generated content adheres to the input prompt. ImageReward(Xu et al., 2023) and PickScore(Kirstain et al., 2023) are employed to assess overall aesthetic quality and perceptual appeal, reflecting general human preferences in visual coherence and composition. Furthermore, we evaluate multidimensional human preferences using MPS(Zhang et al., 2024), a recently proposed metric that captures diverse aspects of human judgment, such as object accuracy, spatial relation, and attribute binding-beyond global similarity. Together, these metrics provide a comprehensive and human-centric evaluation of both fidelity and preference in text-to-image generation. To demonstrate the effectiveness of phase 1 distillation, we compare our 4-step generative models against SDXL(Podell et al., 2023), as well as other open-sourced timestep distillation methods, including LCM-SDXL(Luo et al., 2023a), SDXL-Lighting(Lin et al., 2024), SDXL-Turbo(Sauer et al., 2024b), Realism and Vibrant version of NitroSDChen et al. (2025), and DMD2(Yin et al., 2024b). For phase 2, we further evaluate our approach by comparing it with three reinforcement learning-finetuned models, Hyper-SDXL(Ren et al., 2024), PSO-DMD2Miao et al. (2025), and LPO-SDXLZhang et al. (2025a).

## 4.2  EXPERIMENT ANALYSIS

**Faster and Better Distillation under Phase 1.**    Our method achieves highly efficient distillation from the teacher model, leading to state-of-the-art performance across all benchmarks, by decoupling the distribution matching (DM) and adversarial losses with a timestep-aware strategy, employing PixGAN to alleviate mode-seeking behavior, and stabilizing the generator's score estimator. As shown in Tab. 1, we distill SDXL using various two-time update rules (TTUR), which corresponds to the update frequency ratio between the score estimator and generator. In contrast to DMD2, which uses a TTUR of 5 and thus hinders training efficiency, we experiment with TTUR values of 1, 2, and 5. Our results demonstrate significant improvements in both efficiency and performance. With a TTUR of 5, our model surpasses DMD2 on all benchmarks while requiring only 37.5% of the training cost (batch size × training steps). Reducing the TTUR to 2 allows us to maintain superior human preference scores and comparable text-image consistency with only 8.3% of DMD2's training cost. In the most extreme case, training for only 1,000 steps with a TTUR of 1, with merely 2.1% of DMD2's training cost, we still yield a higher human preference score. Notably, under all tested settings, our model consistently outperforms the original teacher model.

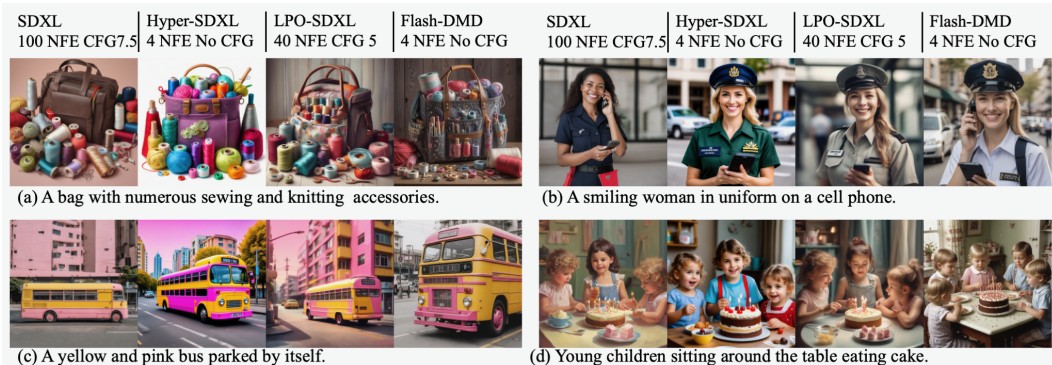

Figure 4: Qualitative comparisons with other reinforcement approaches on SDXL.

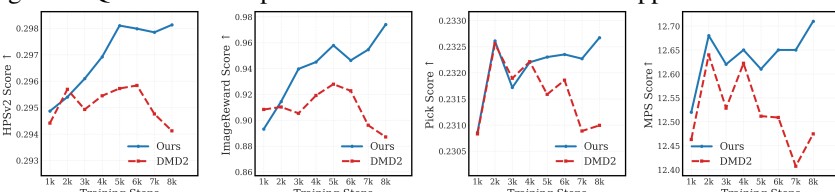

Figure 5: Evaluation results of DMD2 and Flash-DMD (ours) with TTUR=2.

**Boost Performance under Phase 2.** By incorporating reinforcement learning, Flash-DMD achieves a performance comparable to other reinforcement approaches with fewer computational resources, as shown in the Tab. 2 and Fig. 4. Flash-DMDscores the highest on PickScore and MPS, and image fidelity surpasses SDXL and other competitors. Although Hyper-SDXL has the highest ImageReward score and HPSv2 score, it generates overexposed colors and unnatural images. LPO-SDXL gets the highest CLIP score, but it produces oversmoothed images. We speculate that the reason may be that these models use the trained models directly for reinforcement training. This is prone to reward hacking and only rewards the results preferred by the reward model. Meanwhile, Flash-DMDintroduces preference optimization during the training process. With the constraints from distribution matching and PixelGAN, the problem of reward hacking can be alleviated.

## 4.3 ABLATION STUDIES

**Phase1: Comparision with DMD2 under samller TTUR.** We set TTUR=2 for ablate the performance of DMD2 and Flash-DMD under phase 1. The results, presented in Fig. 5, show that our method exhibits stable and continuous improvement throughout the training process. In contrast, DMD2 shows slight initial gains but quickly degrades as training progresses. This comparison validates that our approach offers much better training stability and efficiency under these conditions. To keep the main text focused on core experiments and results, we have moved some detailed ablation studies to the appendix. These additional experiments further validate the design choices of our method and their impact on model performance. Readers can refer to the appendix for more comprehensive data and analysis that support the conclusions presented in the main text.

## 5 CONCLUSION

We present Flash-DMD, a twofold approach that addresses the inefficiencies of existing diffusion distillation methods by leveraging timestep-aware objectives and optimizing the distillation process. In the early phase, Flash-DMDaccelerates convergence by coordinating distribution matching and perceptual realism enhancement. In the later phase, it refines visual details using latent reinforcement learning while preventing mode collapse and artifacts. Experiments show Flash-DMDachieves superior generation quality and the highest human preference scores with significantly reduced training costs. Our method makes diffusion distillation more efficient and accessible, paving the way for advancements in low-step generative modeling. We hope our findings can open new avenues for research and contribute to advancing the field of visual generation.

## ETHICS STATEMENT

All authors affirm that this work adheres to the ICLR Code of Ethics (`https://iclr.cc/public/CodeOfEthics`). We are committed to responsible stewardship of machine learning research and have considered the broader impacts of our work in alignment with the principles of contributing to societal well-being, upholding scientific excellence, avoiding harm, and ensuring fairness. Our method builds upon existing diffusion model frameworks and does not involve direct experimentation with human subjects. All datasets used in this work are established public benchmarks commonly adopted in the generative modeling community, and no new data involving personal or sensitive information has been collected or released. We have taken care to ensure that our approach minimizes potential for misuse and does not introduce new risks related to bias, discrimination, or privacy. The focus of this work is on improving training efficiency and generation quality within a controlled research setting. While advanced generative models may have downstream applications with ethical implications, we emphasize transparency in methodology and encourage future deployment only in socially responsible contexts. We support open and reproducible science while complying with all relevant academic and institutional standards.

## REPRODUCIBILITY STATEMENT

We are committed to ensuring the reproducibility of the results presented in this paper. To this end, we provide detailed descriptions of our method, training procedure, and evaluation protocols in the main text and appendix. We will publicly release both the source code for training and inference, as well as the trained models, to facilitate replication and further research. This open-sourcing plan is intended to support full reproducibility and community engagement.

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

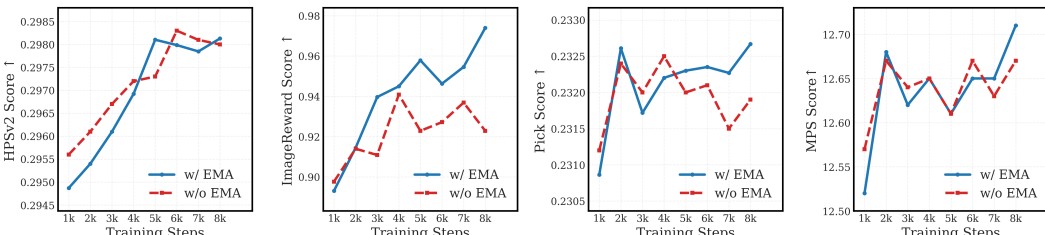

Figure 6: Evaluation results of Flash-DMD (ours) with or without EMA on ImageReward, PickScore, and HPSv2. The training steps range from 1,000 to 8,000. Both models are trained with a two-time scale update rule (TTUR). The generator and the score estimator are updated at a rate of 1:2, i.e., TTUR=2.

# A APPENDIX

## A.1 THE USE OF LARGE LANGUAGE MODELS (LLMS)

In this work, we utilized large language models (LLMs) as assistive tools to refine the writing. Specifically, LLMs were employed to enhance the clarity, coherence, and linguistic quality of these sections without altering the core content or scientific contributions of the paper. The use of LLMs was limited to language refinement and did not play a significant role in research ideation, methodology development, or experimental design. This statement is provided to ensure transparency in accordance with the guidelines for LLM usage.

## A.2 ABLATION STUDIES

**Phase 1: Significance of EMA in Score Estimator.** As described in Sec. 3.3, we employ a Exponential Moving Average (EMA) strategy to help the score estimator more accurately track the generator's distribution, especially under high-frequency updates. To validate the effectiveness of this approach, we conduct an ablation study comparing performance with and without the EMA strategy. As shown in Fig. 6, the model with EMA achieves higher ImageReward and Pickapic scores in later training stages. The HPSv2 scores remain nearly identical. This confirms that the EMA strategy enhances visual quality and human preference without compromising text alignment.

**Phase 2: Trade-Off in Time Scale Update Rule for RL.** Rather than superimposing multiple loss functions via a weighted sum, we use an alternating update strategy to update the generator, applying different loss functions at different frequencies. At the beginning, we initialize the generator and the fake score estimator with weights from TTUR1-1k experiment, and the real score estimator is initialized with SDXL. We trained the model for 2,000 iterations on a single H20 GPU under different frequency ratios between reinforcement loss and distribution matching loss (1:1, 2:1, 5:1, 10:1). The metric comparisons are shown in Table 3. The 5:1 ratio achieves the highest score among these settings.

**Phase 2: Other ablation experiments on reinforcement learning.** *Online training VS Post-training.* We compare online training with post-training by LPO (Zhang et al., 2025b) alone. Our method demonstrates superior performance over Post-Train LPO, which validates the advantage of our proposed training paradigm. and full-noise timestep sampling. *High-noise VS all noise.* Training only on high-noise steps achieves better results than training on all-noise steps. *Including PixelGAN loss.* Furthermore, the incorporation of an additional Pixel-Gan objective yields a positive gain, resulting in a marginal improvement in the metrics. Fig. **??** supplements the results with and without GAN loss across 1,000 to 5,000 iterations. We selected the 5k-step model with GAN loss as our final enhanced version, as it delivered the best performance. The training cost for this model was 12 H20 GPU hours.

Table 3: Comparison of Different Variants in Reinforcement Learning Experiments.

| | ImageReward | PickScore | MPS | CLIP | GPU Hours |
|---|---|---|---|---|---|
| Base-TTUR1-1k | 0.9508 | 0.2322 | 12.672 | 0.3292 | – |
| 1:1 | 0.9135 | 0.2330 | 12.755 | 0.3284 | 5.72 |
| 2:1 | 0.9315 | 0.2329 | 12.770 | 0.3271 | 5.16 |
| 5:1 | 0.9808 | 0.2345 | 12.764 | 0.3275 | 4.83 |
| 10:1 | 0.9640 | 0.2344 | 12.685 | 0.3272 | 4.72 |
| Post-Train LPO | 0.9795 | 0.2345 | 12.689 | 0.3284 | 4.96 |
| all noise | 0.9421 | 0.2331 | 12.800 | 0.3294 | 7.3 |
| + pixelgan | 0.9678 | 0.2345 | 12.812 | 0.3280 | 5.77 |
| Flash-DMD | 1.0004 | 0.2346 | 12.813 | 0.3285 | 12.0 |

Table 4: Mainstream Image Reward Models

| Model | Space | Evaluation Dimension | VLM | Step-aware |
|---|---|---|---|---|
| PickScore | Pixel | Fidelity | CLIP | × |
| ImageReward | Pixel | Fidelity/Aes. | BLIP | × |
| MPS | Pixel | Align./Fidelity/Aes. | CLIP | × |
| HPSv2 | Pixel | Align./Fidelity | CLIP | × |
| SPM | Pixel | Align./Aes. | CLIP | √ |
| LRM | Latent | Align./Aes. | SD1.5/SDXL | √ |
| VisionReward | Pixel | Align./Fidelity | Llama | × |
| UnifiedReward | Pixel | Align. | Qwen2.5VL | × |

### A.3 ANALYSIS OF IMAGE REWARD MODELS

In the Tab. 4, we summarize the scope space and evaluation dimensions of popular image reward models. We also include the Visual Large Model (VLM) used, and whether the timestep is considered for these models.

### A.4 ADDITIONAL QUALITATIVE RESULTS

We show additional qualitative comparison as in Figure7, demonstrating that our model not only surpasses other distillation models but also outperforms the teacher model in refining image quality. Specifically, we compare our results with SDXL, SDXL-Lighting(Lin et al., 2024), SDXL-Turbo(Sauer et al., 2024b), Hyper-SDXL(Ren et al., 2024), DMD2(Yin et al., 2024b), LPO (Zhang et al., 2025b), Realism version of NitroSDChen et al. (2025), PSO(Miao et al., 2025). Additionally, we present qualitative examples of Phase 1-generated images in Figure 8, further highlighting the strengths of our approach.

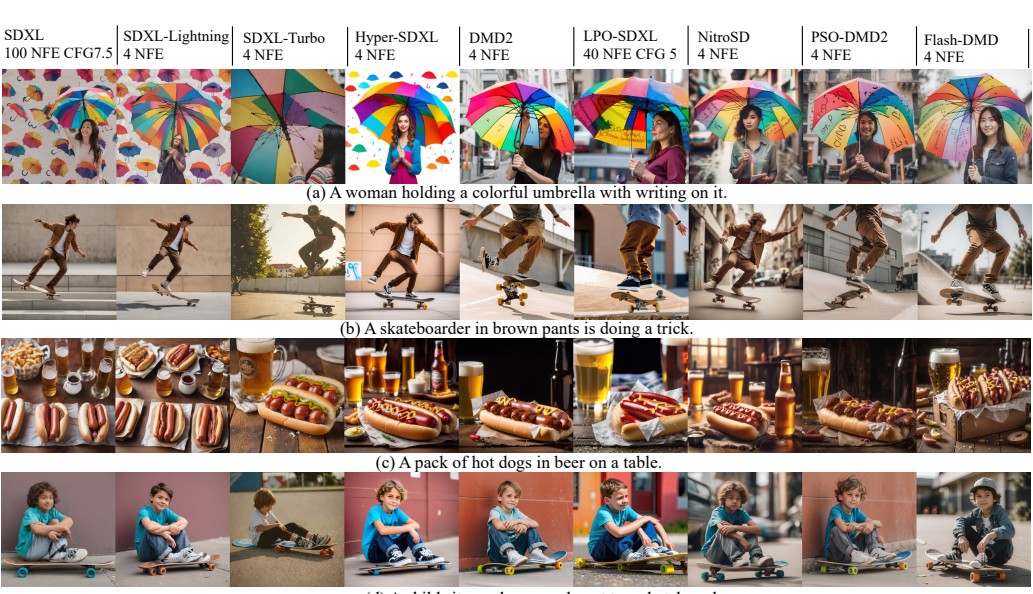

(a) A woman holding a colorful umbrella with writing on it.

(b) A skateboarder in brown pants is doing a trick.

(c) A pack of hot dogs in beer on a table.

(d) A child sits on the ground next to a skateboard.

Figure 7: Qualitative comparisons with other models.

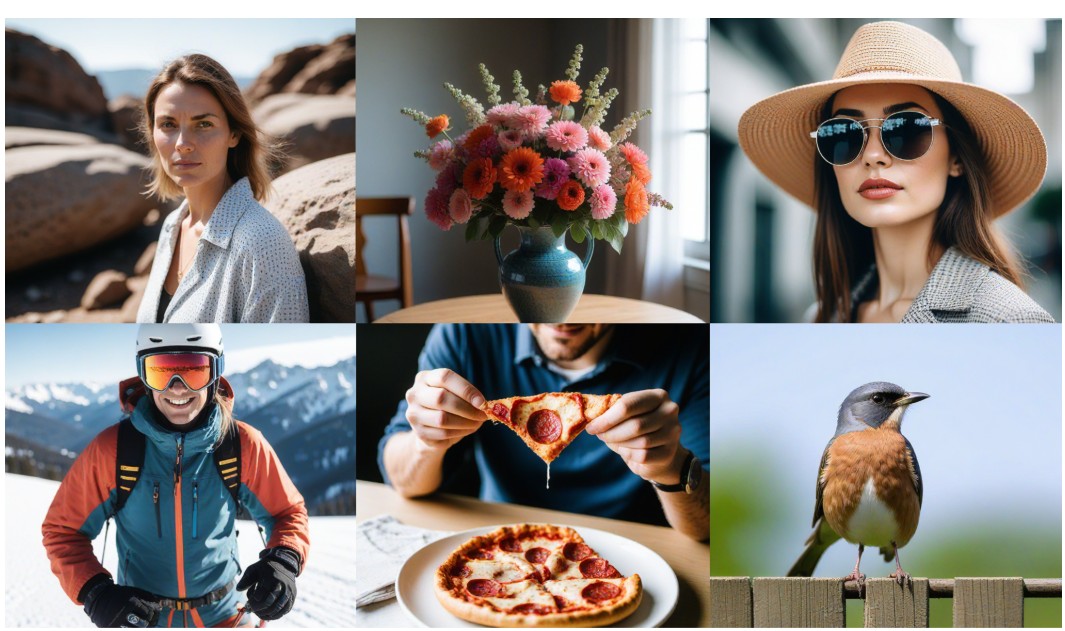

Figure 8: Qualitative results of our first phase model.

