# OpenReview forum: "Flash-DMD: Unifying Distillation and Refinement for High-Fidelity Few-Step Image Generation"
_ICLR.cc/2026/Conference — ICLR 2026 Conference Withdrawn Submission_

### Official Review · Reviewer_yNDc · 2025-10-31

**Soundness:** 4
**Presentation:** 3
**Contribution:** 3
**Rating:** 6
**Confidence:** 3

**Summary:**

This paper proposes flash DMD, a two step approach for training few step generators. The first step is a distillation strategy to learn from the teacher model, while the second step is applying reinforcement learning (through a preference objective) in order to surpass the teacher model in quality.

**Strengths:**

- this paper does a good job intuitively explaining the failures of DMD2, and other methods.
- the overall performance of this method seems pretty strong, and the relative compute amount reduction is significant.

**Weaknesses:**

- I would like to see ablations which separate out the relative gain from the two states. Ie, instead of starting with models trained from the TTUR, instead finetune DMD2 with RL.
- From an RL perspective, the proposed objective seems to be relatively standard, so it is unclear if the benefit comes from here or the LRM.

**Questions:**

- is there a reason to use preferences instead of scores, or is it just because the LRM outputs preferences?
- do we know anything about the generalization of this distillation method. do these techniques work for other domains (eg, molecules).

---

### Official Review · Reviewer_Yvc8 · 2025-11-01

**Soundness:** 2
**Presentation:** 3
**Contribution:** 2
**Rating:** 4
**Confidence:** 5

**Summary:**

This paper combines improved Distribution Matching Distillation and reinforcement learning for SDXL acceleration. The authors analyze the drawbacks of DMD2, and propose Flash-DMD to address these problems. On COCO-10k dataset, experimental results show Flash-DMD achieve higher human preference  scores and text-image alignment than other methods.

**Strengths:**

1. The paper is well-organized and presents its arguments in a clear, coherent manner.
2. This paper conducts an in-depth analysis of DMD2, thereby uncovering the training inefficiency issue inherent in the DMD series. To address this, it introduces a timestep-aware strategy for model distillation and a SAM-based Pixel-GAN to enhance realism and structural coherence.
3. Flash-DMD uses significantly reduces the training hours, yielding higher human preference scores and comparable text-image consistency than DMD2-SDXL, SDXL-Lightning.

**Weaknesses:**

1. The strategy to improve the training efficiency of DMD series is tricky, the novelty is limited.

2. FID, a commonly used metric is not reported in the Table. Authors need to add FID to Table 1 and 2.

3. This paper adopts COCO-10k dataset to evaluate all methods,  but DMD2 uses COCO-2014 30K. For a fair comparison, the author need to present the results on COCO-2014 30K for all methods.

4. Does Flash-DMD support different sampling steps via a unified model. The performance for 1,2, 8 sampling steps is missing. If Flash-DMD only supports 4-step sampler, it will limit the flexibily in the application.  According to Table 1, 2, it seems that reforcement learning only introduces marginal improvement witht respect to human preference metrics, and even leads to lower CLIP score

5. This paper only validates the effectiveness of Flash-DMD on SDXL. Can it be used to accelerate rectified flow models, such as SD3 or Flex-dev.1. How about its performance on rectified flow models.

**Questions:**

1. The application of Pixel-GAN will siginificantly increase the consumption of GPU memory, how do authors address it.

2. The author should cite PCM[1] and compare with it.

Reference

1.Phased Consistency Model.

---

### Official Review · Reviewer_VXu9 · 2025-11-02

**Soundness:** 3
**Presentation:** 3
**Contribution:** 2
**Rating:** 6
**Confidence:** 3

**Summary:**

This paper proposes a novel framework that addresses key limitations of existing diffusion models—high computational costs in sampling and training and instability in reinforcement learning (RL)-based refinement—while enabling high-fidelity few-step (4-step) image generation.

In the fast distillation phase, it uses a timestep-aware strategy: DM loss aligns the student model with the teacher’s global structure at high-noise timesteps, while adversarial loss enhances detail and realism at low-noise steps. A SAM-based Pixel-GAN mitigates mode-seeking, and a stabilized score estimator (with EMA updates) cuts training costs—achieving better human preference scores than DMD2 with only 2.1–37.5% of its training cost.

In the RL refinement phase, Flash-DMD integrates a Latent Reward Model (LRM) to optimize visual quality. It samples high-noise latents for diverse "win-lose" image pairs, using RL loss to refine details, while ongoing distillation constraints prevent artifacts. This phase maintains realism and avoids over-smoothing, outperforming RL-fine-tuned models like Hyper-SDXL with far lower GPU hours (12 H20 vs. 400 A100).

Experiments on COCO-10k confirm Flash-DMD’s superiority in CLIP scores, aesthetic metrics (ImageReward, PickScore), and human preference (MPS), making efficient, high-fidelity diffusion models more accessible. Codes and models will be open-sourced for reproducibility.

**Strengths:**

1. Flash-DMD drastically reduces training costs compared to baseline methods like DMD2. It achieves state-of-the-art human preference scores with only 2.1%–37.5% of DMD2’s training cost.

2. By decoupling distribution matching (DM) loss and adversarial loss across timesteps—using DM loss for global structure at high noise and adversarial loss for fine details at low noise—it avoids the conflicting objectives that plague DMD2.

3. It is rigorously evaluated across multiple metrics (CLIP, ImageReward, PickScore, MPS) on the COCO-10k dataset, outperforming both distillation baselines (SDXL-Turbo, DMD2) and RL-finetuned models. The authors also provide ablation studies (e.g., EMA impact, timestep sampling) and ethical/reproducibility commitments (open-sourced code/models), strengthening credibility.

**Weaknesses:**

1. Flash-DMD is optimized for 4-step generation, and its performance in scenarios requiring fewer (1-step) or more (e.g., 10-step) steps is not thoroughly explored. Its timestep-aware strategy may need reconfiguration for other step counts, limiting flexibility.

2. While short-term training stability is demonstrated (e.g., no degradation in 8,000 steps), the paper does not evaluate long-term training dynamics (e.g., over 50,000 steps) or potential drift in the student model’s alignment with the teacher, leaving uncertainty about scalability for large-scale applications.

**Questions:**

NA

---

### Official Review · Reviewer_hAoW · 2025-11-06

**Soundness:** 2
**Presentation:** 2
**Contribution:** 2
**Rating:** 2
**Confidence:** 4

**Summary:**

This paper proposes Flash-DMD, a few-step distillation method, which addresses training instability issues in Distribution Matching Distillation (DMD2), specifically 4-step DMD2, and reduces the overall training cost. The paper hypothesizes that the misalignment between distribution matching loss objective and adversarial loss objective in DMD2 results in poor training convergence and proposes to decouple these two objectives. Flash-DMD operates in two phases. In the first phase, Flash-DMD applies distribution matching (DM) loss to high-noise timesteps (t=999, 749, 499) to establish global structure and adversarial loss to low-noise timesteps (t=249) to refine textures. In the second phase, along with DM objective, the paper also integrates RLHF objective via alternate Two-time update rule (TTUR). The human preference is optimized at only high-noise timesteps (t=999, 749). The paper argues that the distillation process acts as regularizer and prevents reward hacking which is common in post-training preference approaches. The resulting method achieved competitive performance with human-preference metrics on COCO-10K Text-to-image benchmark.

**Strengths:**

1. Performance: The proposed method outperforms teacher model and also uses a fraction of compute necessary for DMD2. As shown in Table 1, Flash-DMD needs around ~16.7% - 37.5% of DMD2’s training cost to achieve comparable or better metrics.
2. Motivation: The paper identifies the source of training inefficiency in DMD-v2 and proposes heuristics to minimize the source of training in-efficiency.
3. Architectural improvements: This paper uses a discriminator based on the Segment Anything Model (SAM) to enforce realism in pixel space.

**Weaknesses:**

1. The method is very much tailored to 4-step DMD2. It is unclear if these advantages transfer to multiple steps or even to other distribution matching based distillation approaches. This paper picks a specific DMD2 and uses extensive hyper parameter tuning and heuristics to optimize its performance. The overall method is not very theoretically grounded.
2. Writing: Some sections of the paper have multiple typos and need to be rewritten to improve clarity. Examples of such sections are Section 3.4 on reinforcement learning. There are also missing details such as the  details of hyperparameter $\beta$ used in preference optimization loss. Sentences such as line 325-326 “The dynamic threshold is removed” are unclear. The training details including hyperparameters should be specified.
3. Missing ablations: The paper claims that PixelGAN alleviates mode-seeking behavior but this hasn’t been empirically shown with an ablation.

**Questions:**

1. There’s no theory that shows that Latent consistency models (LCMs) follow PF-ODE. Further, LCMs don’t necessarily perform well with 4 sampling steps. The decision of using LCM as a sampler for the reward modeling seems a bit arbitrary. Why is LCM a reasonable choice?

---

### Note · Authors · 2025-11-14

I have read and agree with the venue's withdrawal policy on behalf of myself and my co-authors.